# Acrylamide Exposure from Common Culinary Preparations in Spain, in Household, Catering and Industrial Settings

**DOI:** 10.3390/foods10092008

**Published:** 2021-08-26

**Authors:** Lucía González-Mulero, Marta Mesías, Francisco J. Morales, Cristina Delgado-Andrade

**Affiliations:** Institute of Food Science, Technology and Nutrition (ICTAN-CSIC), 28040 Madrid, Spain; l.gonzalez@ictan.csic.es (L.G.-M.); fjmorales@ictan.csic.es (F.J.M.); cdelgado@ictan.csic.es (C.D.-A.)

**Keywords:** acrylamide, exposure, Maillard reaction, culinary processes, catering, industry, households

## Abstract

In 2019, the European Commission recommended monitoring the presence of acrylamide in certain foods not included in Regulation 2158/2017, to consider other sources of exposure to the contaminant. In the present study, eleven groups of processed foods commonly consumed in Spain were classified, according to their food matrix, into potato-based food, cereal-based food and food based on cereal mixed with meat, fish or vegetables. Samples were collected from three different settings: household, catering services and industrial origin, to evaluate the influence of the food preparation site on acrylamide formation. The highest concentrations of acrylamide were observed in chips (French fries), especially those prepared at home. Although at lower levels, all the other foods also contained significant concentrations of acrylamide, confirming the need to control its content in foods not included in the EU regulation. Industrially processed foods made a lower contribution to acrylamide exposure, probably due to the more stringent controls exercised on culinary processes in this context. The higher levels recorded for households and catering services highlight the need for greater awareness of culinary processes and for measures to be adopted in these settings to limit the formation of acrylamide in food preparation.

## 1. Introduction

In recent years, economic and sociocultural factors have produced significant changes in eating habits in developed countries. Urban lifestyles, changes in the distribution of household tasks, a lack of time and scant interest among the younger generations in learning traditional forms of cooking have all contributed to a greater reliance on caterers, company canteens, institutional catering, restaurants and other food supply services [1]. Moreover, ‘fast food’ options have become firmly established among certain population groups, at the expense of fresh food [2]. This shift in dietary habits also involves a change in the profile of foods consumed, resulting in a greater consumption of precooked products, to be quickly and easily prepared or finished at home. According to a 2020 government report on food consumption in Spain [3], the consumption of ready-cooked dishes increased by 11.3% since 2019. In many cases, these food preparations are based on a protein source combined with cereals or potato. This type of food composition, when subjected to common culinary operations such as roasting, toasting, baking, or frying, can result in the development of the Maillard reaction within the food matrix and hence the formation of acrylamide [4,5].

Acrylamide is a process chemical contaminant. Its presence in food consumption increases the risk of developing cancer in all age groups [4]. Dietary exposure to acrylamide is a cross-sectional event, as it is contained in many types of foods, regardless of the area of production (industrial, household or catering services). In the last decade, the food industry has implemented strategies to reduce its impact, especially in the products most likely to generate critical levels of acrylamide. In this respect, the Confederation of the Food and Drink Industries of the EU (later renamed FoodDrinkEurope) developed its “Acrylamide Toolbox” to harmonise mitigation strategies in the food processing industry [6]. In 2017, in a related initiative, the European Commission established mitigation measures and benchmarks to reduce the presence of acrylamide in foods, focusing in particular on the industrial processing of three main sources of acrylamide: potato-based products, cereal derivatives and coffee and its substitutes [7]. However, the large variety of foods consumed by European populations makes it necessary to consider other food matrixes as possible sources of this contaminant. In this sense, previous researchers identified foodstuffs such as biscuits and bread derivatives [8], roasted chestnut products [9] or roasted coca beans (dark chocolate) [10] as a source of acrylamide. Accordingly, in 2019, the European Commission issued further recommendations to monitor the presence of acrylamide in other potato-based products (croquettes, potato casserole, potato/meat dishes, etc.), bakery products (croissants, doughnuts, pancakes, etc.), cereal products (rice and maize crackers, cereal snacks, etc.) and other foods (dried fruits, olives in brine, vegetable crisps, etc.) [11], in order to consider the adoption of possible risk management measures, complementing those already provided by Regulation 2158/2017.

The great variability of culinary elaborations in domestic and industrial settings and in catering services is directly reflected in the acrylamide content of foods, and hence the impact produced on consumers. This variability arises from differences in raw materials, recipes, processing conditions and decisions made by the operator/consumer, among other aspects. In the domestic context, for example, according to the degree of bread toasting employed, acrylamide levels can vary by up to 8%, while certain conditions of potato frying can increase the total dietary exposure to acrylamide by up to 80% [4]. Consequently, in order to establish the contribution of a given culinary preparation to acrylamide exposure, it is necessary to consider various recipes for the same (or similar) end product being prepared by different persons within different settings, or diverse commercial brands in the case of marketed products. A previous observational study conducted in 208 Spanish households revealed the great variability in acrylamide formation for a process as apparently simple as potato frying [12]. Therefore, in culinary preparations with more complex recipes and processes, even greater variability is to be expected.

In view of these considerations, and in the light of European Food Safety Authority (EFSA) requirements, together with the multiple scenarios in which acrylamide exposure can take place, it seems urgently necessary to obtain and publish reliable exposure data not only for typical dietary sources of acrylamide but also for complex culinary preparations, including precursors. Accordingly, the main aim of the present study is to establish the range of acrylamide exposure from common culinary preparations within Spanish diets, considering household, industrial and catering settings, and taking into account recent changes in patterns of food consumption.

## 2. Materials and Methods

### 2.1. Chemicals and Reagents

Acrylamide standard (99%), potassium hexacyanoferrate (II) trihydrate (98%, Carrez-I) and zinc acetate dihydrate (>99%, Carrez-II) were obtained from Sigma (St. Louis, MO, USA). Acrylamide labelled by ^13^C_3_ (99%) was acquired from Cambridge Isotope Laboratories (Andover, MA, USA). Formic acid (98%), methanol (99.5%) and hexane were purchased from Panreac (Barcelona, Spain). Deionised water was produced by a Milli-Q Integral 5 water purification system (Millipore, Billerica, MA, USA). All other chemicals, solvents and reagents were of analytical grade. Oasis-HLB cartridges (30 mg, 1 mL) were obtained from Waters (Milford, MA, USA). Cellulose syringe filter units (0.22 µm) were purchased from Análisis Vínicos (Tomelloso, Ciudad Real, Spain).

### 2.2. Sample Design

Eleven types of processed foods that are characteristically consumed in Spain were selected for analysis. The samples were divided into three groups, by main food types, as follows: (1) potato-based foods (chips (French fries); Spanish potato omelette); (2) cereal-based food (‘*torrijas*’; sponge cake); (3) foods based on cereal mix with meat, fish or vegetables (breaded fillet; ham and cheese fillet; pizza; meat, tuna and/or vegetable puff pastry pies; meat, tuna and/or vegetable patties; ‘*migas*’, croquettes).

The dishes were prepared and collected from three different settings: (i) a domestic setting: five volunteers prepared the foods at home. Each participant performed the culinary procedure following their usual method, with no prior operational restriction in terms of the selection of raw materials, the cooking conditions and techniques employed; (ii) a catering setting: the same culinary preparations of the selected foods were acquired in collective catering and takeaway establishments; (iii) a food industry setting: ready-to-eat or precooked preparations of the selected items were purchased in different supermarkets. Five samples of different commercial brands of each food group were considered. For foods that were not ready-to-eat, the cooking instructions supplied regarding technique (mainly baking and frying) and intensity (time, temperature or power) were strictly followed. Samples were collected and analysed from January 2021 to April 2021. Appendix A provides a detailed description of the ingredients and culinary techniques applied in each case. Samples were coded, photographed and weighed, and the edible fraction of each serving portion was taken, weighed and homogenised with a hand blender (Taurus, Vital CM, Spain). The samples were then stored at −20 °C until analysis.

### 2.3. Acrylamide Determination by Liquid Chromatography–Electrospray Ionisation–Tandem Mass Spectrometry (LC-MS/MS)

For each sample, the acrylamide content was determined as follows, in accordance with Mesias and Morales [13], with some modifications. Two grams of ground samples were weighed and mixed with 37.6 mL of Milli-Q water in polypropylene centrifugal tubes. A total of 4 mL of hexane was added to the tubes to remove the fat content of the foods. All the samples were spiked with 400 μL of a 5 μg/mL (^13^C_3_)-acrylamide methanolic solution as an internal standard and later homogenised (Ultra Turrax, IKA, Mod-T10 basic, Bohn, Germany) for 15 min. The sample was then treated with 1 mL of Carrez I (15 g of potassium ferrocyanide/100 mL of water) and 1 mL of Carrez II (30 g of zinc acetate/100 mL of water) solutions and centrifuged (9000× *g* for 10 min) at 4 °C. The hexane was removed, and the samples were cleaned using Oasis-HLB cartridges, preconditioned with 1 mL of methanol and 1 mL of water. An aliquot of the clear supernatant (1 mL) was loaded into the cartridge at a flow rate of 2 mL/min; the first drops were discarded, after which the remainder were collected. The solution was then filtered through a 0.22 μm filter into an amberlite LC-MS vial.

Sample extracts and calibration standards were analysed on an Agilent 1200 liquid chromatograph coupled to an Agilent Triple Quadrupole MS detector (Agilent Technologies, Palo Alto, CA, USA). Analytical separation was achieved with an Inertsil ODS-3 column (250 × 4.6 mm, 5 μm; GL Sciences Inc., Tokyo, Japan) at 30 °C. Isocratic elution was achieved with a mobile phase of formic acid in water (0.2 mL/100 mL) at a flow rate of 0.4 mL/min. The injection volume was 10 μL. Electrospray ionisation in the positive ionisation mode was used. Under these chromatographic conditions, acrylamide eluted at 6.1 min. The needle was set at 1.0 kV. Nitrogen was used as the nebulizer gas (12.0 L/min) and the source temperature was set at 350 °C. Signals at *m*/*z* 72.1–*m*/*z* 55.1 and *m*/*z* 75.1–*m*/*z* 58.1 were isolated for acrylamide and (^13^C_3_)-acrylamide, respectively. For the transitions *m*/*z* 72.1 > *m*/*z* 55.1 and *m*/*z* 75.1 > *m*/*z* 58.1, the fragmentation was set at 76 V and the collision energy at 8 V. Masses were recorded using multiple reactions monitoring (MRM). For quantitation, the signals at *m*/*z* 75.1 and 78.1 were used, while signals at *m*/*z* 58.1 and 55.1 served for qualification.

The recovery rate of acrylamide spiked to the samples ranged between 94–106% for potato, between 90–105% for cereal-based food and between 90–102% for food based on cereal mixed with meat, fish or vegetables. The precision, repeatability and reproducibility of the analytical method were evaluated by analysing different samples on the same day (precision), by different operators (repeatability) and on different days (reproducibility). The relative standard deviations (RSDs) of the analysis were 2.8%, 1.2% and 2.5% for the precision, repeatability and reproducibility, respectively. The limit of detection (LOD) and limit of quantification (LOQ) were calculated by injecting lower concentrations of standards. A concentration determined as a signal-to-noise ratio of 3 was assigned to LOD (4.5 µg/kg), and that determined as a signal-to-noise ratio of 10 was assigned to LOQ (15 μg/kg). All analyses were performed twice. The results for acrylamide are expressed as μg/kg of the sample.

### 2.4. Estimation of Dietary Exposure to Acrylamide

Dietary exposure to acrylamide was estimated by considering the average serving size for each dish (g/serving) multiplied by the mean acrylamide content obtained for all categories of processed foods (µg/kg). The three preparation settings (households, catering services and industrial) were first compared within each food analysed and then globally, including all the foods, in order to analyse the variations in exposure according to the preparation scenario. When the level of acrylamide in a given dish was non-detectable (<15 µg/kg), the results for the samples in question were replaced by numerical values corresponding to half of the reported LOQ, in line with WHO recommendations [14].

### 2.5. Statistical Analyses

All statistical analyses were performed using SPSS version 26 (SPSS, Chicago, IL, USA). Data are expressed as the mean ± standard deviation (SD). One-way ANOVA, followed by Scheffe’s test or Student’s *t*-test, was used to determine the overall significance of the differences obtained. Variance homogeneity was determined by Levene’s test. All statistical parameters were evaluated assuming a level of significance of *p* < 0.05.

## 3. Results and Discussion

Certain foods commonly consumed in Spain were selected for analysis. The selection criteria included food preferences, contemporary lifestyles and the culinary preparations routinely employed in the three settings considered as possible scenarios for acrylamide exposure (households, catering services and industrial origin). Some of these foods are usually consumed during specific periods or in limited geographical areas. For instance, *torrijas* is a sweet dish typically consumed at Easter, while *migas* is a traditional rural dish made basically with bread and other ingredients, which vary from one region to another. Nevertheless, minor differences in the composition of a given type of food are not relevant to our study aims, and so only the standard (or most common) form of culinary preparation was considered. All the food preparations selected for analysis contained a protein source in their composition together with cereals or potato, thus providing acrylamide precursors. In addition, the culinary operations applied in all of them promote the formation of the contaminant. Appendix A describes the ingredients of the food preparations considered and the cooking methods used in their elaboration.

### 3.1. Acrylamide Content in the Food Groups

Table 1 shows the acrylamide content in each of the samples analysed. The chips, by a significant margin, presented the highest average acrylamide content (391 µg/kg). However, lower values for this food have been reported elsewhere by the EFSA [4], and in studies of samples obtained from households [15] and from local markets [16]. On the other hand, higher mean values have been found in samples from restaurants [15]. These previously reported findings are summarised in Table 2. Chips also presented the greatest variability in acrylamide content, ranging from 23 to 1114 µg/kg, a pattern that is in line with data reported previously (Table 2). Our own research group also documented this huge variability in the household setting (27–4200 µg/kg) [12], in catering services (<20–1068 µg/kg) [5] and in primary school canteens (20–4000 µg/kg) [17]. Sanny et al. [18] reported similar results for restaurants (152–1023 µg/kg), institutional caterers (151–505 µg/kg) and chain fast food services (150–392 µg/kg). In the other samples considered, acrylamide contents were lower, with the mean values ranging from 21 µg/kg (*torrijas*) to 78 µg/kg (Spanish omelette). Thus, chips represent an outlier in this food dataset, a finding that should be acknowledged and evaluated in statistical processing. If chips were excluded from the global evaluation, the acrylamide values obtained for the Spanish omelette would be significantly higher (*p* < 0.05) than those for all other food groups, with contents ranging from 23 to 183 µg/kg. These data are of the same order of magnitude as those reported by Bermudo et al. [19] (135 µg/kg), Delgado-Andrade et al. [20] (128 µg/kg) and Branciari et al. [21] (151 µg/kg), but lower than those of Delgado-Andrade et al. [22] (240 µg/kg) (Table 2). These results show that among the food groups sampled, those based on potatoes are the main contributors to the total exposure to dietary acrylamide [4]. This effect is due to the high level of reducing sugars and free asparagine in the fresh tuber, on the one hand, and to their method of preparation (frying), on the other [23,24]. Although both products are based on processed potatoes, the levels of acrylamide in chips and Spanish omelette are substantially different, probably due to the different moisture levels in the sources of potato used in each case. For the Spanish omelette, potatoes are submitted to a light frying process and then mixed with egg prior to the preparation of this composite food. In contrast, chips are more intensely fried, a process that provokes more severe dehydration. The egg content in the dish could also be associated with the lower acrylamide levels measured in Spanish omelette, as eggs are rich in sulphur-containing amino acids [25]. These molecules have nucleophilic side chains that can originate Michael-type addition reactions with acrylamide from cooked potatoes, which decrease the free acrylamide content in the final product [26].

Among the other food groups analysed, there were no significant differences in the mean acrylamide values (*p* > 0.05) (Table 1). To calculate these mean values for samples with acrylamide levels below the LOQ (15 µg/kg), a level of 7.5 µg acrylamide/kg was assumed. The acrylamide levels obtained for sponge cakes (<LOQ-138 µg/kg), pizzas (<LOQ-104 µg/kg), patties (<LOQ-131 µg/kg) and *migas* (<LOQ-95 µg/kg) were comparable. Further down the scale, the levels for *torrijas* (<LOQ–58 µg/kg) were similar to those of breaded fillets (<LOQ-64 µg/kg), ham and cheese fillets (<LOQ-48 µg/kg), puff pastry pies (<LOQ-66 µg/kg) and croquettes (<LOQ-45 µg/kg). These results are in line with those reported previously (Table 2). In agreement with these concentrations, Esposito et al. [36] recently reported that the mean and median levels of acrylamide in several cereal-derived baked foods collected at local manufacturers of Campania region (Italy), were above the benchmark levels issued by EU regulation 2017/2158.

The low acrylamide content observed in all food groups except chips and Spanish omelette could be attributed to the food matrix composition and the culinary technique or processes used in their preparation. Thus, the low acrylamide concentration detected in the *torrijas* (mean value 21 µg/kg) could be due to the type of bread used, the frying conditions or even to the high moisture content of this food, since the bread is immersed in milk for some minutes prior to frying. In the case of sponge cake, the different proportions of ingredients and differences in baking temperatures and times could explain the variability in the acrylamide levels detected. For pizza, puff pastry pies and patties, it might be significant that acrylamide is formed mainly in the dough used for their preparation, while the ingredients included in the filling would make a lower contribution, with the exception of some processed vegetables, which already contain the contaminant due to their previous processing. Differences among these types of foods could be attributed to the different proportions of dough in the final product or to differences in the processing techniques applied (baking or frying). Similarly, the formation of acrylamide in coated fried/roasted foods such as breaded fillet, ham and cheese fillet and croquettes mainly takes place in the crust and will depend on the composition of the coating. The type of coating material and its composition, including breadcrumbs or flour, with or without the addition of water, milk, egg, salt, potato starch and seasonings, and shortenings, amongst other ingredients [37], contribute to determining the levels of reducing sugars and asparagine in the product. In this respect, Esposito et al. [36] concluded that it seems necessary to better understand the role of sugars, eggs, butter and other ingredients in the formation mechanism of acrylamide during the production of biscuits and other sweets, in order to improve the recipes from the perspective of new mitigation approaches. The level of acrylamide-forming precursors and again the culinary technique applied (roasting or frying) definitively influences the formation of acrylamide. The maximum concentration reached was similar for these three food groups. Finally, the type of bread used to make the *migas* and the heat treatment applied influences the formation of acrylamide. Before preparing this dish, the bread is moistened or immersed in water for a few moments, which delays the development of browning when the *migas* are sautéed in the pan. However, the contribution of the other ingredients, especially peppers, should also be considered. For these products, too, acrylamide occurrence after processing has been reported [38], as in the case of other heat-treated vegetables [5].

### 3.2. Effect of the Culinary Setting

The influence of the culinary setting on acrylamide content in the different groups of processed foods was considered by examining three scenarios: households, industrial preparation and catering services. Figure 1 shows the distribution of the samples analysed, by type of food. 

The highest acrylamide values were found in the home-prepared chips, with values ranging from 71 µg/kg to 1114 µg/kg, which is in line with previous reports in this respect, according to which in the domestic setting, and depending on the potato frying conditions, the total dietary exposure to acrylamide could be increased by up to 80% [4,29]. By contrast, the lowest acrylamide levels in our samples were recorded for the chips prepared in the industrial setting (23–656 µg/kg). The variations observed can be explained by differences in frying conditions, in culinary techniques and in the characteristics of the potatoes used. Take-away establishments and collective catering services frequently use commercial par-fried frozen potatoes, which are prepared in a procedure (blanching) that reduces the level of reducing sugars [39]. This circumstance could account for the low level of acrylamide observed in chips produced in industrial settings. Nevertheless, five samples presented an acrylamide content exceeding the European Regulation benchmark value of 500 µg/kg [7] in the three settings (Figure 1). 

For the remaining food groups, no benchmark levels for acrylamide content have yet been established, although some of these products are addressed by European Commission Recommendation 2019/1888, which presents a non-exhaustive list of foods to be monitored [11]. These foods varied widely in acrylamide content, according to the culinary setting considered, as shown in Figure 1. Overall, none of these cases included foods with the highest levels of acrylamide, the highest concentrations of which were found in *torrijas* (58 µg/kg), ham and cheese fillet (48 µg/kg) and *migas* (95 µg/kg) for the household environment; in Spanish omelette (183 µg/kg), sponge cake (138 µg/kg), breaded fillet (64 µg/kg), pizza (104 µg/kg) and croquettes (45 µg/kg) for catering services; and in puff pastry pies (56 µg/kg) and patties (131 µg/kg) for the foods of industrial origin. The fact that many of the samples obtained from the catering setting presented the highest levels of acrylamide may be indicative of a lack of control during food preparation and suggests there is a need to standardise the variety of practices employed by food handlers [18,40].

For the Spanish omelette samples, statistically significant differences were detected between the different settings, with a higher acrylamide content in the catering setting than in the household and industrial contexts (*p* < 0.05). The levels recorded are within the range of concentrations observed by Delgado-Andrade et al. [20] and Branciari et al. [21] in Spanish omelettes prepared by catering services (Table 2). The same pattern was observed for the breaded fillet, where despite the wide range of values among the samples, significant differences in terms of the preparation context were only observed between catering services and the other two settings. Similar results have been reported by Mesías et al. [33] (Table 2). For the *torrijas* group, the household samples presented significantly higher acrylamide levels than those of industrial origin, very likely due to the less rigorous control exercised during the cooking process.

The generally lower levels of acrylamide measured in most of the samples obtained from the industrial setting suggests there is a stricter control of food preparation in this sector, including attention to the raw materials used, their handling and cooking. This scrupulous approach might also reflect the impact of the control measures established by European regulation 2017/2158 [7], a consequence that has also been observed in the industrial production of crisps (potato chips). In this respect, Powers et al. [41] corroborated the effectiveness of the mitigation measures applied in crisp manufacturing, observing a downward trend in average levels of acrylamide, from 763 µg/kg in 2002 to 372 µg/kg in 2019, according to the European Snacks Association. This period was characterised by falls not only in the maximum values of acrylamide, but also in the percentage of samples exceeding the reference value for this food (750 µg/kg). This reduction in acrylamide levels in crisps appears to be due to stricter attention to the temperature and duration of the cooking, blanching before frying (the removal of sugars and other soluble metabolites in hot water), vacuum frying, the control of moisture levels in the final product, the selection of appropriate varieties of fresh potatoes, according to their end use, and the careful control of the storage temperature and conditions of the fresh materials, as well as post-frying quality control by reference to the colour of the food [42]. All of these mitigation measures, together with other recommendations for the food production sector, are compiled and described in the Acrylamide Toolbox [6]. 

### 3.3. Acrylamide Exposure in Different Food Groups and Settings

The acrylamide content per serving is valuable information that could influence consumers’ dietary choices. Table 3 shows the mean values and ranges of acrylamide exposure per serving, assuming a normal portion size for each type of food. Our study data show that the consumption of chips produces the highest exposure to this contaminant, especially when they are prepared at home (mean value: 53.8 µg/serving). These findings are in agreement with the acrylamide levels assumed by Grob et al. [43], for whom average levels of exposure were 20–70 µg for a 100 g serving. The frequent consumption of chips, especially by the teenage population, makes this food one of the main dietary sources of acrylamide [21]. Other important foods in this respect include Spanish omelette (18.5 µg/serving) and patties (13.2 µg/serving). Both of these values were obtained from foods prepared by catering services, which contribute significantly to the dietary exposure to acrylamide, more so than those prepared in household and industrial settings. On the other hand, lower levels of acrylamide exposure from the consumption of Spanish omelette were reported by Delgado-Andrade et al. [20,22] (12 and 6.4 µg/serving in samples obtained from institutional catering and school canteens, respectively). At a lower scale, the same situation was observed in the mean exposure values for breaded fillets (4.3 µg/serving in food prepared by catering services vs. 1.5 and 1.2 µg/serving for household and industrial settings, respectively) and pizza (5.0 µg/serving for catering services vs. 2.9 µg/serving for households and 2.8 µg/serving for industrial settings). The lowest acrylamide exposure in food prepared by catering services was recorded for croquettes (1.5 µg/serving) compared to other breaded samples such as breaded fillets (4.3 µg/serving) and ham and cheese fillet (3.4 µg/serving). By preparation context, significant differences (*p* < 0.05) were only observed between catering services and the other two settings for the Spanish omelette and the breaded fillet, and between household and industrial settings for the *torrijas* (Table 3). Considerable variability was observed in the results for the three settings, with five outliers in catering services, four in household preparation and three in the industrial setting (Figure 2).

In summary, food preparation in the catering sector contributed most to acrylamide exposure in most of the food groups analysed, with the exception of chips, *torrijas* and *migas,* where the mean exposure was greater in the domestic preparations, and puff pastry pies, where exposure was highest in the industrial samples. These results underscore the need to control acrylamide levels in foods beyond those listed in the European regulation, as possible contributors to acrylamide exposure in the diet, as indicated in the 2019/1888 recommendations issued by the European Commission [11]. Overall, the differences observed are not statistically significant (*p* > 0.05), but industrially processed foods tend to make a lower contribution to acrylamide exposure than those produced in household and catering settings (Figure 2), very probably due to the application of mitigation strategies and/or stricter process control in the industrial setting. The need for these measures is emphasised in the Acrylamide Toolbox, together with the recommendation that a risk/benefit analysis should be conducted, considering the balance between the nutritional quality and the acrylamide levels present in foods [6].

## 4. Conclusions

This study highlights the variability in acrylamide levels in different food preparations commonly consumed in Spain. The highest levels of this contaminant were observed in chips (French fries), which confirms that this food is one of the principal sources of dietary exposure to acrylamide. Although at lower levels, acrylamide was also present in the other food samples considered, which underscores the need to control its content in foods not yet addressed in EU food quality regulations. The wide variability observed in acrylamide concentration might be explained by differences in the composition of raw materials, in the manufacturing process, in culinary preparation techniques and in the recipes used, within each of the food preparation settings analysed. The lowest acrylamide values were observed in industrially processed foods, in comparison to catering services and food preparation in the household. By standard portion size, chips produce the highest exposure to acrylamide, especially when they are prepared in the domestic environment. Although the differences are not statistically significant, industrially processed foods contribute less to acrylamide exposure than those prepared in the household and by catering services. These results probably reflect the existence of stricter control of the culinary process in the industrial environment, possibly associated with the implementation of the mitigation measurements required by the European regulation concerning this sector. Accordingly, stricter monitoring is needed of food cooking processes and mitigation measures should be promoted to control acrylamide formation in foods prepared in the household and by catering services. These actions will help reduce the amounts of acrylamide in processed foods and alleviate the risk associated with exposure to this contaminant from common culinary preparations in Spain.

## Figures and Tables

**Figure 1 foods-10-02008-f001:**
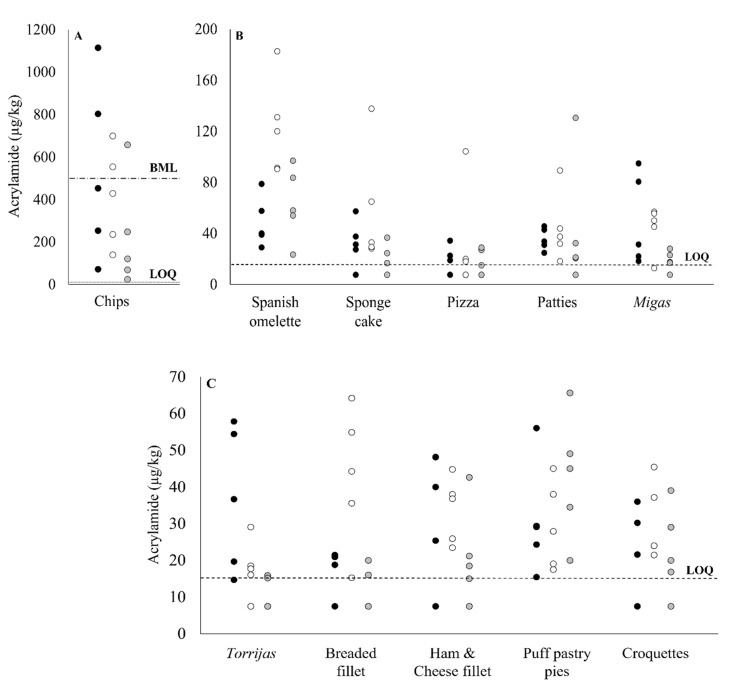
Acrylamide content (µg/kg) (ranges 0-1200 µg/kg (**A**), 0-200 µg/kg (**B**) and 0-70 µg/kg (**C**)) in the groups of processed foods prepared in each of the three settings. Black dots represent the household setting, white dots collective catering and grey dots industrial origin. BML: benchmark level established for chips by the European Regulation 2158/2017 (500 µg/kg). LOQ: limit of quantification.

**Figure 2 foods-10-02008-f002:**
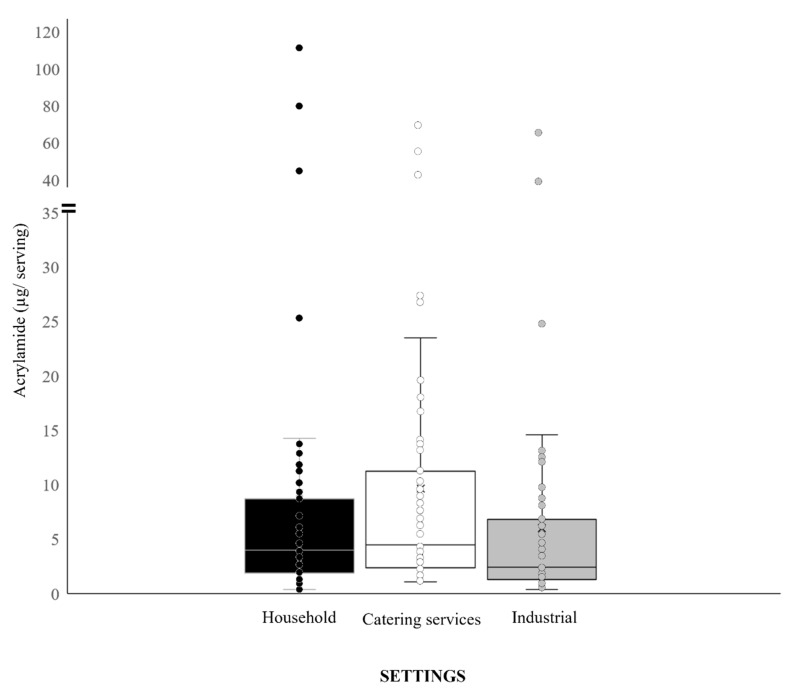
Distribution of acrylamide exposure (µg/serving) in the three settings (household, catering services and industrial) with each of the processed food groups analysed (chips, Spanish omelette, *torrijas*, sponge cake, breaded fillet, ham and cheese fillet, pizza, puff pastry pie, patties, *migas*, croquettes). Values are mean ± SD.

**Table 1 foods-10-02008-t001:** Acrylamide content (µg/kg) in eleven groups of processed foods representing foods commonly consumed by the Spanish population.

Food Group	% Samples <LOQ	Acrylamide (µg/kg)					
		Mean	SD	Median	P90	Minimum	Maximum
Potato-based food							
Chips	-	391a	320	253	927	23	1114
Spanish omelette	-	78b	43	79	152	23	183
Cereal-based food							
*Torrijas*	33	21b	16	16	56	<LOQ	58
Sponge cake	20	36b	32	29	94	<LOQ	138
Food based on cereal mix with meat, fish or vegetables							
Breaded fillet	33	23b	18	19	59	<LOQ	64
Ham and cheese fillet	20	27b	14	25	46	<LOQ	48
Pizza	40	22b	24	18	62	<LOQ	104
Puff pastry pies	-	34b	15	29	60	15	66
Patties	7	41b	31	32	106	<LOQ	131
*Migas*	7	37b	26	28	86	<LOQ	95
Croquettes	20	24b	12	24	42	<LOQ	45

Results represent the mean (n = 15), standard deviation (SD), median, 90th percentile (P90), minimum and maximum levels. LOQ: limit of quantification (15 µg/kg). Values below the LOQ were set at half the LOQ (7.5 µg/kg). Different letters mean significant differences between the food groups (*p* < 0.05).

**Table 2 foods-10-02008-t002:** Acrylamide concentration (µg/kg) reported by different international studies for the food groups analysed in the present study.

Food Groups	Acrylamide (µg/kg)	Analytical Methodology	Origin	Literature Reference
Potato-based food				
Chips	63–799	GC-MS/MS	Restaurant	[27]
	152–1023	GC-TOF-MS	Restaurant	[18]
	151–505	GC-TOF-MS	Institutional caterers	[18]
	150–392	GC-TOF-MS	Chain fast food service	[18]
	291–970	LC-ESI-MS/MS	Restaurant	[28]
	1590–2390	LC-ESI-MS/MS	Household	[28]
	201	n.i.	Data submitted by food associations	[4]
	332	n.i.	Data submitted by European countries	[4]
	20–1330	n.i.	Household	[29]
	190–1090	n.i.	Restaurant	[29]
	418	LC-MS/MS	Restaurant	[15]
	206	LC-MS/MS	Household	[15]
	27–4200	LC-ESI-MS/MS	Household	[12]
	20–4000	LC-ESI-MS/MS	School canteen	[17]
	187	LC-ESI-MS	Local markets	[16]
	36–1411	HPLC-DAD	Street vendors	[30]
Spanish omelette	135	LC-MS/MS	n.i.	[19]
	240 ^a^	LC-ESI-MS	Supermarkets	[22]
	128 ^a^	LC-ESI-MS	Catering	[20]
	151	HPLC-DAD	School canteen	[21]
Cereal-based food				
*Torrijas*	16 ^a^	LC-ESI-MS	Collective catering	[20]
Sponge cake	<30	LC-MS/MS	Supermarkets	[31]
Food based on cereal mix with meat, fish or vegetables				
Breaded fillet	19	n.i.	n.i.	[32]
	40	LC-ESI-MS/MS	Restaurant/school canteen	[33]
	451	GC-MS	Local supplier (market)	[34]
Ham and cheese fillet	32	LC-ESI-MS/MS	Restaurant/school canteen	[33]
Croquettes	285–420	RP-HPLC	Supermarkets	[35]
	36	LC-ESI-MS/MS	Restaurant/school canteen	[33]

n.i.: not indicated. ^a^ Measured in kg of dry matter. GC-TOF-MS: gas chromatography time-of-flight mass spectrometry; LC-ESI-MS/MS: liquid chromatography electrospray ionisation tandem mass spectrometry; GC-MS/MS: gas chromatography tandem mass spectrometry; LC-MS/MS: liquid chromatography tandem mass spectrometry; HPLC-DAD: high-performance liquid chromatography coupled with diode array detection; RP-HPLC: reversed-phase high-performance liquid chromatography.

**Table 3 foods-10-02008-t003:** Estimation of acrylamide exposure from different food groups and settings.

Food Group	Portion (g)	Exposure (µg/Serving)					
		Household		Catering Services		Industrial	
		Mean ± SD	Min–Max	Mean ± SD	Min–Max	Mean ± SD	Min–Max
Potato-based food							
Chips	100	53.8 ± 42.1a	7.1–111.4	41.1 ± 22.7a	13.9–69.8	22.3 ± 25.6a	2.3–65.6
Spanish omelette	150	7.3 ± 2.9a	4.5–12.2	18.5 ± 5.7b	14.0–28.3	9.5 ± 4.3a	4.6–15.0
Cereal-based food							
*Torrijas*	160	5.9 ± 3.1a	3.2–9.5	2.8 ± 1.2ab	2.5–4.8	2.1 ± 0.6b	1.3–2.6
Sponge cake	60	1.9 ± 1.1a	1.7–3.6	3.5 ± 2.8a	1.8–4.1	1.1 ± 0.7a	1.1–2.3
Food based on cereal mix with meat, fish or vegetables							
Breaded fillet	100	1.5 ± 0.7a	1.9–2.1	4.3 ± 1.9b	1.5–6.4	1.2 ± 0.6a	1.6–2.0
Ham and cheese fillet	100	2.6 ± 1.9a	2.5–4.8	3.4 ± 0.9a	2.3–4.5	2.1 ± 1.3a	1.5–4.3
Pizza	160	2.9 ± 1.8a	3.0–5.5	5.0 ± 6.6a	2.9–16.7	2.8 ± 1.7a	2.4–4.6
Puff pastry pies	180	6.2 ± 3.0a	2.8–10.2	5.9 ± 2.4a	3.2–8.2	8.6 ± 3.4a	3.7–12.0
Patties	310	10.7 ± 2.6a	7.7–14.1	13.2 ± 8.1a	5.6–27.7	12.7 ± 15.0a	6.4–40.5
*Migas*	150	7.4 ± 5.3a	2.8–15.1	6.6 ± 2.7a	2.5–8.9	2.8 ± 1.2a	2.6–4.4
Croquettes	50	1.0 ± 0.7a	1.2–2.0	1.5 ± 0.5a	1.2–2.6	1.1 ± 0.6a	0.9–2.2

SD: Standard deviation. Different letters in the same row mean significant differences (*p* < 0.05) between the settings.

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
