# Peer review of "Acrylamide Exposure from Common Culinary Preparations in Spain, in Household, Catering and Industrial Settings"

_foods, 2021, doi:10.3390/foods10092008_

Round 1

Reviewer 1 Report

The authors have done a very original research, harmonizing with the needs of the European Commission and all those interested in the subject of the presence of acrylamide in food.

The research was well planned and executed. I am pleased with the professional presentation of the results and the good discussion. Conclusions arise from the data obtained and their discussion, and in my opinion are very important in the light of the continuous evaluation of the exposure of acrylamide intake with the diet.

The authors often cited their previous studies but these citations are needed and justified. Moreover, they confirm the familiarity of the entire research team with the topic addressed. Therefore, I recommend the manuscript after minor revision. Below is a list of my observations and comments.

Introduction

Perhaps it is worth for raising the reader's awareness to mention in the introduction that various foodstuffs (not only those studied by the authors) can be a source of acrylamide, which is proven by previous research:

  1. Impact of Raw Materials and Production Processes on Furan and Acrylamide Contents in Dark Chocolate, https://dx.doi.org/10.1021/acs.jafc.0c00412
  2. Acrylamide content of selected Spanish foods: Survey of biscuits and bread derivatives, https://doi.org/10.1080/02652030601101169
  3. Determination of acrylamide in roasted chestnuts and chestnut-based foods by isotope dilution HPLC-MS/MS, https://doi.org/10.1016/j.foodchem.2008.11.057

Materials and methods

line 121: please provide what kind of material syringe filter was made (PTFE, nylon, other?)

line 151: instead “removed” will be better “taken”

line 162-163: Was an internal standard added to all samples?

line 169-171: What solvents were used to wash the samples applied to the SPE columns and what was used to elute the acrylamide from the columns? Please add this information.

line 173: please provide the solution volume subjected to filtration on syringe filter

line 174-175: please add the table with recovery rates given separately for each group of foods analyzed in your research

It is necessary and it will be also great improvement of the article if you provide informations about analysis conditions and devices used to determine acrylamide by LC-MS/MS method. This informations is missing in the article.

Please provide LOD value for the method and extend the procedure of determination of precision, repeatability and reproducibility of the analysis.

Tables and figure are not numbered chronologically, causing great confusion for the reader. Please change the numbers.

Results and discussion

lines 218-222: entire sentence needs to be rewritten to make it clearer

Errors! – line 227, 275 –please correct

Line 275-277: perhaps it would be more correct to take the LOD value for samples where the acrylamide level is below the LOQ of the method. I give this to the authors for consideration.

Figure 1 caption

please provide the better explanation (from the text) of BML abbreviation and give the value of BML

Author Response

Answer to the comments point by point

Reviewer 1

The authors have done a very original research, harmonizing with the needs of the European Commission and all those interested in the subject of the presence of acrylamide in food.

The research was well planned and executed. I am pleased with the professional presentation of the results and the good discussion. Conclusions arise from the data obtained and their discussion, and in my opinion are very important in the light of the continuous evaluation of the exposure of acrylamide intake with the diet.

The authors often cited their previous studies but these citations are needed and justified. Moreover, they confirm the familiarity of the entire research team with the topic addressed. Therefore, I recommend the manuscript after minor revision. Below is a list of my observations and comments.

Thank you very much for your comments.

Introduction

Perhaps it is worth for raising the reader's awareness to mention in the introduction that various foodstuffs (not only those studied by the authors) can be a source of acrylamide, which is proven by previous research:

  1. Impact of Raw Materials and Production Processes on Furan and Acrylamide Contents in Dark Chocolate, https://dx.doi.org/10.1021/acs.jafc.0c00412
  2. Acrylamide content of selected Spanish foods: Survey of biscuits and bread derivatives, https://doi.org/10.1080/02652030601101169
  3. Determination of acrylamide in roasted chestnuts and chestnut-based foods by isotope dilution HPLC-MS/MS, https://doi.org/10.1016/j.foodchem.2008.11.057

Suggestion has been considered and modifications have been included in the introduction of the revised manuscript (lines 71-74). Articles have been mentioned (references 8-10) and, consequently, the rest of the references have been renumbered.

Materials and methods

line 121: please provide what kind of material syringe filter was made (PTFE, nylon, other?)

Cellullose syringe filter were used. This aspect has been mentioned in line 124.

line 151: instead “removed” will be better “taken”

Change has been performed (line 154).

line 162-163: Was an internal standard added to all samples?

All samples were added with the internal standard. This aspect has been specified in line 165: “All the samples were spiked with 400 μL of a 5 μg/mL [13C3]-acrylamide methanolic solution as an internal standard and later homogenised (Ultra Turrax, IKA, Mod-T10 basic, Germany) for 15 min”.

line 169-171: What solvents were used to wash the samples applied to the SPE columns and what was used to elute the acrylamide from the columns? Please add this information.

Oasis-HLB cartridges were preconditioned with 1 ml of methanol and 1 ml of water. After that 1 mL of the clear supernatant was loaded into the cartridge at a flow rate of 2 mL/min; the first drops were discarded, after which the remainder were collected. This aspect is mentioned in lines 172-176.

Oasis HLB cartridges contain the Oasis HLB sorbent, which is a universal polymeric reversed-phase sorbent developed for the extraction of a wide range of acidic, basic, and neutral compounds from various matrices using a simple, generic protocol. In this case, compounds that may cause interferences are retained during filtration of the samples, but acrylamide is not retained. For that reason, it is not necessary to use solvents for its elution and, consequently, this aspect has not mentioned in the manuscript.

line 173: please provide the solution volume subjected to filtration on syringe filter

This aspect is mentioned in lines 173-174: “An aliquot of the clear supernatant (1 mL) was loaded into the cartridge”.

line 174-175: please add the table with recovery rates given separately for each group of foods analyzed in your research

Recovery rates for each group of foods analysed have been given separately. If referee agrees, authors consider a better option to mention these numerical data within the manuscript (lines 197-199).

It is necessary and it will be also great improvement of the article if you provide informations about analysis conditions and devices used to determine acrylamide by LC-MS/MS method. This informations is missing in the article.

A new paragraph specifying the conditions for analysis of acrylamide has been included in the revised manuscript (lines 178-195).

Please provide LOD value for the method and extend the procedure of determination of precision, repeatability and reproducibility of the analysis.

Following the referee’s comment, this information has been extended in the revised version of the manuscript (lines 199-210).

Tables and figure are not numbered chronologically, causing great confusion for the reader. Please change the numbers.

Thanks for the comments. This aspect has been modified, references to Table 3 and Figure 2 have been removed from material and methods in order to avoid confusions (lines 217-221).

Results and discussion

lines 218-222: entire sentence needs to be rewritten to make it clearer

Sentence has been rewritten (lines 249-253).

Errors! – line 227, 275 –please correct

The authors consider these sentences an error in the conversion of the manuscript to PDF format. The conversion of the revised manuscript will be carefully checked to avoid these mistakes.

Line 275-277: perhaps it would be more correct to take the LOD value for samples where the acrylamide level is below the LOQ of the method. I give this to the authors for consideration.

If referee agrees, authors consider keeping the selected option following the WHO recommendations, which indicate that those samples with non-detectable levels of acrylamide should be replaced by the numerical value corresponding to half of the reported LOQ [reference 14].

Figure 1 caption

please provide the better explanation (from the text) of BML abbreviation and give the value of BML

Figure 1 caption has been modified in order to include both the value and a better explanation of BML.

Reviewer 2 Report

The manuscript is an interesting paper dealing with acrylamide exposure from common culinary Spanish preparations.

Some suggestions:

Please correct the error reference in section 3.1

Authors could consider the following paper to enhance the discussions on occurrence and dietary exposure through typical foods from Italian diet:

Esposito et al. (2020), Molecules 25(18), 4156, 2020

Author Response

Answer to the comments point by point

Reviewer 2

The manuscript is an interesting paper dealing with acrylamide exposure from common culinary Spanish preparations.

Thank you very much for your comments.

Some suggestions:

Please correct the error reference in section 3.1

Authors apologize for the mistake and appreciate the referee's observation. The number of references has been corrected.

Authors could consider the following paper to enhance the discussions on occurrence and dietary exposure through typical foods from Italian diet:

Esposito et al. (2020), Molecules 25(18), 4156, 2020

Authors have considered the referee’s suggestion and this new reference has been mentioned in the revised manuscript (lines 315-319 and 346-350).
